# Evaluating the impact of maternal health care policy on stillbirth and perinatal mortality in Ghana; a mixed method approach using two rounds of Ghana demographic and health survey data sets and qualitative design technique

**John Azaare** [1,2]*, **Patricia Akweongo**[2], **Genevieve Cecilia Aryeteey**[2], **Duah Dwomoh**[3]

1 Department of Health Services, Policy Planning, Management and Economic, School of Public Health, University for Development Studies, Tamale, Ghana, 2 Department of Health Policy, Planning and Management, School of Public Health, University of Ghana, Legon, Accra, Ghana, 3 Department of Biostatistics, School Public Health, University of Ghana, Legon, Accra, Ghana

* jazaarejn@uds.edu.gh

**Data Availability Statement:** The quantitative data sets used in the current analysis are freely available

## Abstract

### Background

Stillbirth and perinatal mortality issues continue to receive inadequate policy attention in Ghana despite government efforts maternal health care policy intervention over the years. The development has raised concerns as to whether Ghana can achieve the World Health Organization target of 12 per 1000 live births by the year 2030.

### Purpose

In this study, we compared stillbirth and perinatal mortality between two groups of women who registered and benefitted from Ghana's 'free' maternal health care policy and those who did not. We further explored the contextual factors of utilization of maternal health care under the 'free' policy to find explanation to the quantitative findings.

### Methods

The study adopted a mixed method approach, first using two rounds of Ghana Demographic and Health Survey data sets, 2008 and 2014 as baseline and end line respectively. We constructed outcome variables of stillbirth and perinatal mortality from the under 5 mortality variables (n = 487). We then analyzed for association using multiple logistics regression and checked for sensitivity and over dispersion using Poisson and negative binomial regression models, while adjusting for confounding. We also conducted 23 in-depth interviews and 8 focus group discussions for doctors, midwives and pregnant women and analyzed the contents of the transcripts thematically with verbatim quotes.

to the public from the DHS Programme website https://dhsprogram.com/data/. However, the qualitative data was collected from human subjects to triangulate the quantitative data, and cannot be publicly shared. The qualitative data is available from hppm@ug.edu.gh upon request.

**Funding:** The authors received no specific funding for this work.

**Competing interests:** The authors have declared that no competing interest exist.

## Results

Stillbirth rate increased in 2014 by 2 per 1000 live births. On the other hand, perinatal mortality rate declined within the same period by 4 per 1000 live births. Newborns were 1.64 times more likely to be stillborn; aOR: 1.64; 95% [CI: 1.02, 2.65] and 2.04 times more likely to die before their 6th day of life; aOR: 2.04; 95% [CI: 1.28, 3.25] among the 'free' maternal health care policy group, compared to the no 'free' maternal health care policy group, and the differences were statistically significant, p< 0.041; p< 0.003, respectively. Routine medicines such as folic acid and multi-vitamins were intermittently in short supply forcing private purchase by pregnant women to augment their routine requirement. Also, pregnant women in labor took in local concoction as oxytocin, ostensibly to fast track the labor process and inadvertently leading to complications of uterine rapture thus, increasing the risk of stillbirths.

## Conclusion

Even though perinatal mortality rate declined overall in 2014, the proportion of stillbirth and perinatal death is declining slowly despite the 'free' policy intervention. Shortage of medicine commodities, inadequate monitoring of labor process coupled with pregnant women intake of traditional herbs, perhaps explains the current rate of stillbirth and perinatal death.

## Introduction

Stillbirth, defined as the expulsion of a fetus with no sign of life remain a global issue of public health concern [1, 2]. In practice, gestational age of 20 to 28 weeks or a birth weight of 350 to 1000g is usually required to determine stillbirth [1]. According to the World Health Organization (WHO) stillbirth is a neglected tragedy which poses economic, social and psychological effects to families, especially, the affected mother often leading to social withdrawal, loneliness and depression [3–5].

Globally, 1 in 72 babies are stillborn and this translate to nearly 2 million stillbirths annually [4, 6]. Although the current estimates represent a global reduction of 35% overall since 2000, the rate of reduction is thought to be inadequate, and hence, the World would not achieve its target of 12/1000 live births by 2030 [4, 5, 7]. Of the global estimates, 84% is accounted for by lower and middle-income countries particularly, sub-Saharan Africa and southern Asia where three of four global stillbirth estimates are reported [4, 8, 9].

In sub-Saharan Africa, stillbirth increased from 0.77 million in 2000 to 0.82 million in 2019, representing a 42 percent global stillbirths and, suggesting that the Africa region can only attain the 12 per 1000 lives birth target in 2050 [4, 10]. Averagely, 21.7 per 1000 babies were stillborn in Ghana in 2019 and although this is an improvement over previous years, it is yet high compared to the regional average [11, 12]. Stillbirths remain a critical indicator of maternal and child health care performance and reflect negatively on weak health systems, particularly in lower and middle-income countries [7, 13, 14].

Recently, the WHO launched an action plan campaign for newborn care, Every Newborn Child Action Plan (ENAP) to re-vitalize efforts toward reducing global stillbirths with emphasis on access to quality maternal health care as a necessary means to achieving country-level targets [15]. Although the ENAP is yet to receive the required attention [6, 16], the country-level programme of 'free' maternal health care policy was introduced with the object of driving access to maternal health care, and ultimately improving newborn care outcomes [17, 18].

In line with the access to care policy, Ghana declared the 'free' registration of pregnant women as an exemption package of its national health insurance scheme (NHIS) in 2008 aimed at bridging the access gap to care to improve utilization of maternal health care services, thereby mitigating inequalities effect to enhance newborn care survival [19, 20].

While the 'free' policy targeted access to maternal health care, in particular, its broader intentions included comprehensive caregiving of newborns up to 90 days post-delivery [21, 22]. The 'free' maternal health care policy (FMHCP) initiative received £42.5 in funding support from the then UK and has since served over 3 million beneficiaries since its inception [22, 23]. Although the 'free' policy lacked an implementation framework at its inception, it nonetheless gained popularity, as pregnant women received cost 'free' maternal health care at a cost to the NHIS [23–25].

In this paper, we compare stillbirth outcomes between mothers who registered and benefitted from the 'free' maternal health care policy since its inception and mothers who did not. Given the quintessence of health system factors effect on stillbirth [26], we further explored the views of service providers and pregnant women from selected health facilities to add context to the quantitative results to inform our discussion and conclusions.

## Conceptual framework

We hypothesized that certain factors undermine the successful operationalization of the 'free' maternal health care policy in Ghana (Fig 1) and hinder the intention of the policy and its ability to bring about a decline in not just maternal mortality, but also stillbirth and perinatal mortality in the medium to long term. The 'free' maternal health care policy is administered via the NHIS which in itself is bedeviled with funding constraints culminating in delays in payment of claims over the years [27, 28].

Consequently, it appears the existing challenges affect the effective management of accredited service provider facilities thereby threatening the credibility of the purchaser-provider split concept [29]. On another level, the Ghanaian society presents itself as a keeper of pregnant women with cultural demands that upset the effective implementation of the 'free' policy. These unintended practices seem to derail the efforts of health care professionals and policymakers and hence, frustrate their efforts of achieving reduced mortality outcomes of newborn care.

Studies have shown that maternal age, rural/urban area of residence, twin pregnancy, negative pregnancy outcome, income level, education and marital status play roles to moderate the outcome of stillbirth and perinatal mortality [30–32]. In this current study, we ask to what extent does the FMHCP intervention affect the outcome of stillbirth and perinatal mortality against the background of the policy implementation bottleneck while accounting for the moderating factors?

## Contextual definition

**Impact:** Reduction in stillbirth and perinatal mortality over time between 2008 and 2014.
 **Stillbirth:** The birth of a fetus after 28 weeks gestation with no signs of life.
 **Perinatal mortality:** The death of newborns within 6 days of life.
 **'Free' maternal health care policy:** Pregnant women registrants of the national health insurance scheme (NHIS).

## Materials and methods

### Study design

The current study employed a sequential mixed method design: first, analyzing repeated cross-section data of Ghana Demographic and Health Survey (GDHS), 2008 and 2014 as baseline

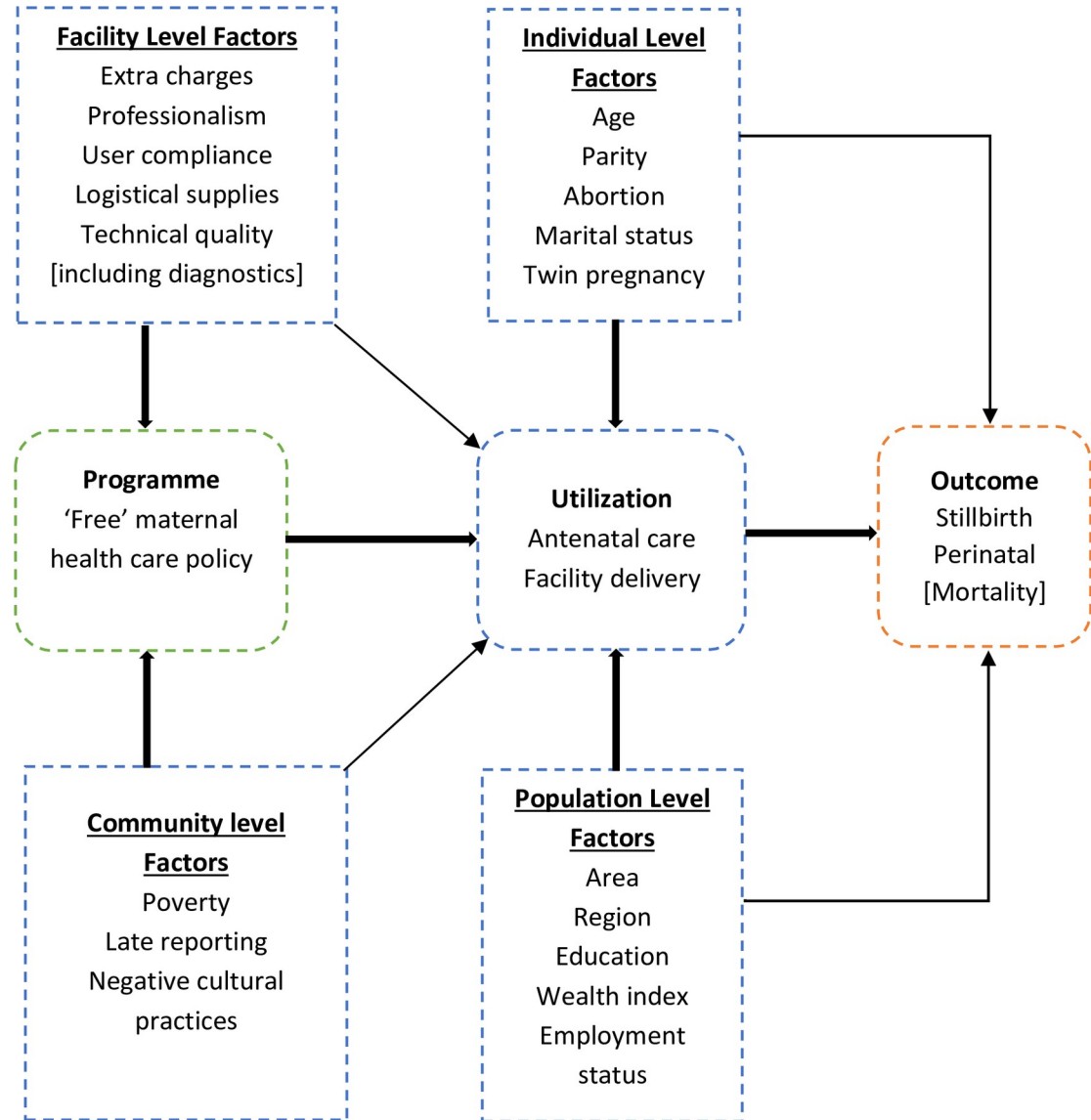

**Fig 1. Conceptual framework: Multifaceted factors moderating stillbirth and perinatal mortality.**

and end line, respectively. The two-rounds of DHS data sets were merged, and two groups created: the 'free' maternal health care policy group and the 'no-free' maternal health care policy group. We then used the NHIS registration status of women as a proxy for our exposure variable (the 'free' maternal health care policy) and constructed stillbirth and perinatal death as our outcome variables using the under 5 mortalities variable from the merged data sets.

The first author conducted one-on-one in-depth interviews with 5 medical doctors and 18 midwives at post to the antenatal clinics and labor units of the selected health facilities and conducted 8 focus group discussions (FGD) for pregnant women who accessed care using the 'free' policy in the selected hospitals, to obtain user perspective for a contextual understanding of health system factors that affect stillbirth outcomes. Given the time disparity of the secondary data and qualitative interviews, facility-level data from two regions of Ghana, obtained from the District Health Information Management Systems (DHIMS) [33] was analyzed and

triangulated with those of the regression output from the GDHS to provide a current undertone to the outcomes of stillbirth and perinatal death for a meaningful inference.

## Qualitative study setting

The Upper East and Northern regions (Fig 2) are among the poorest regions of Ghana. The Upper East region has a total land area of about 8,842sq km with Bolgatanga as its capital and an estimated population of approximately 1.5 million [34, 35]. The region has a national health insurance enrollment rate of 6.3% of its total population with a considerably good number of midwives compared to other regions of Ghana [21]. Between 2016 and 2020, stillbirth figures increased in trend as per the regional data from DHIMS (Fig 3).

On the other hand, the Northern region shared a boundary with the Upper East Region at the time of the DHS data collection (now with the North East region) with a relatively low literacy rate [36]. The Northern region has a land mass of about 70,765.2km$^2$ with an estimated population of about 1.8m representing 9.6% of the total population of Ghana and shows a considerably stable but inconsistent decline in stillbirth proportionate to facility delivery utilization per the DHIMS record (Table 1, Fig 3).

## Study participants, sampling and variable construction

**Quantitative.** Stillbirth and perinatal death variables were generated from the under 5 mortality variables using STATA 15 to construct binary outcomes of '1' representing

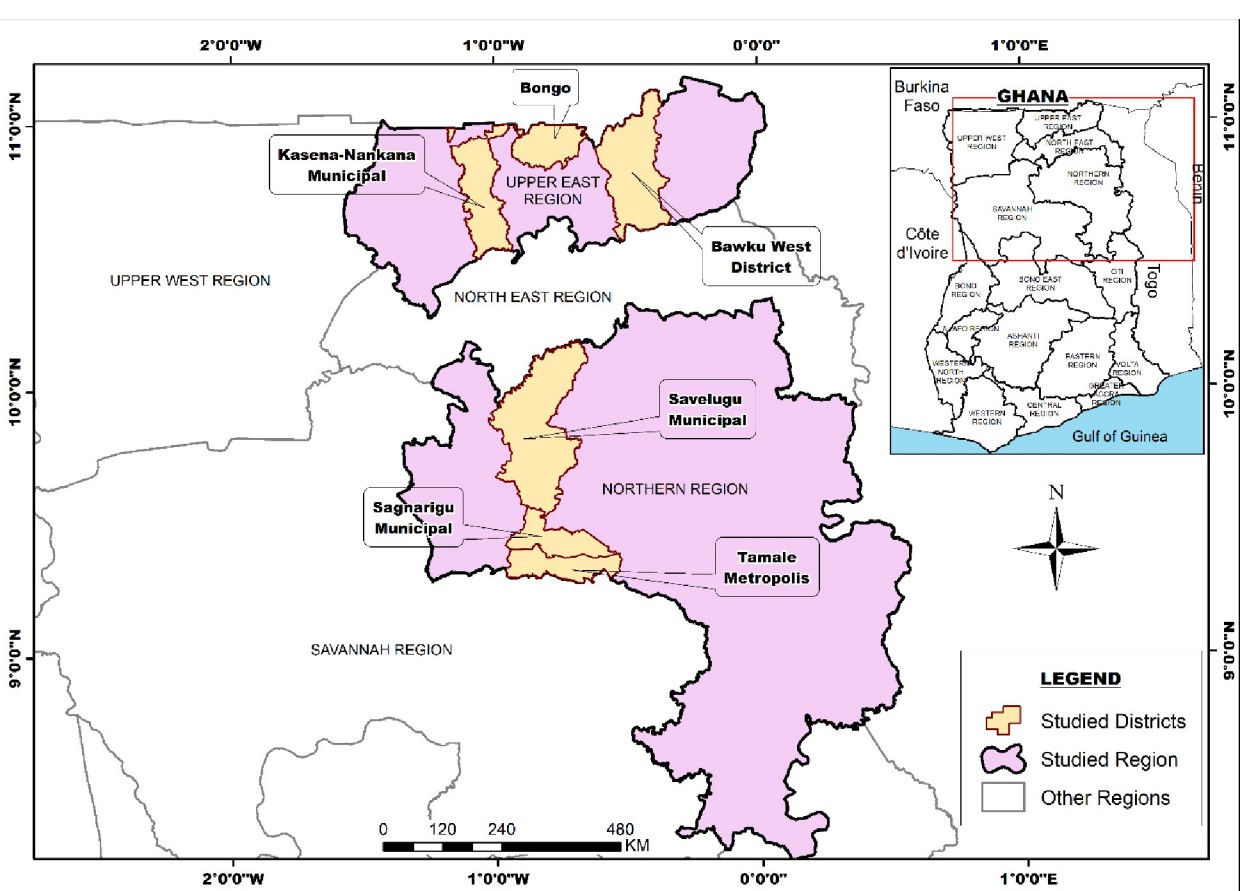

**Fig 2. Map of the Upper East and Northern regions of Ghana, showing selected districts of study.**

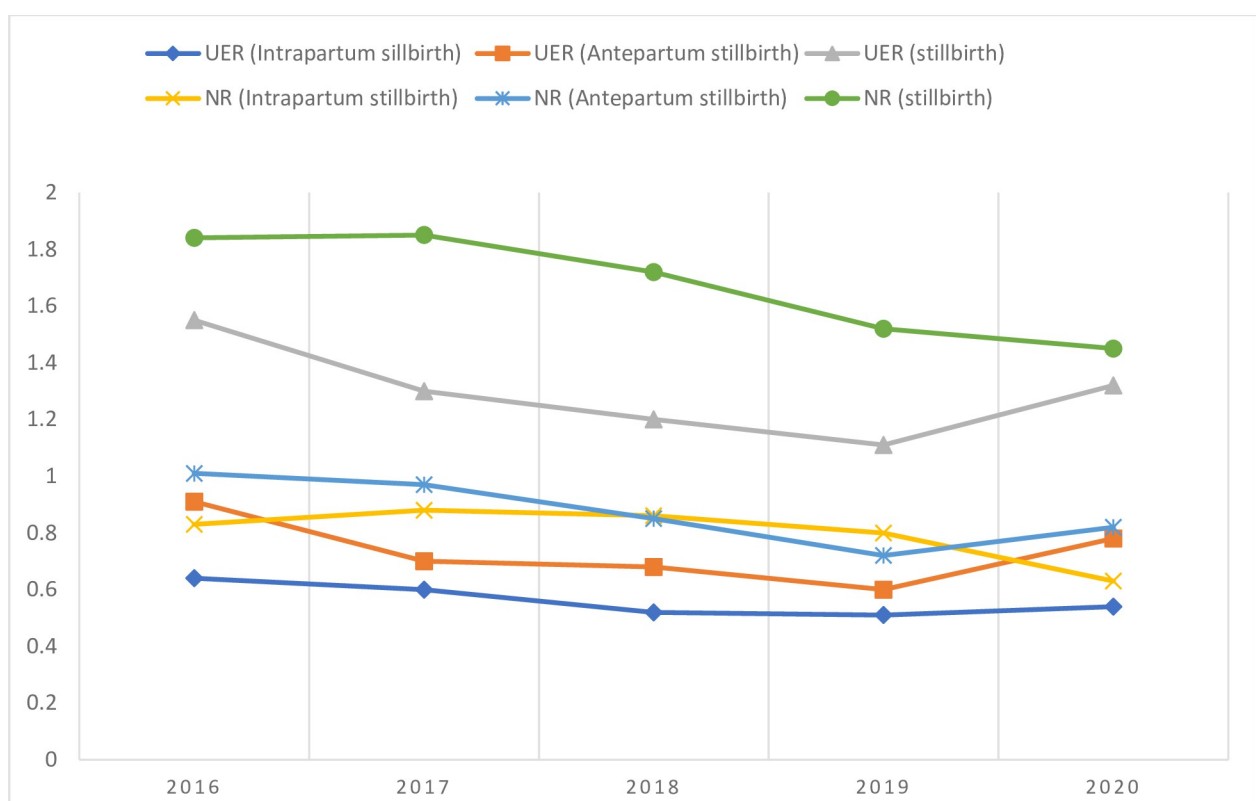

**Fig 3. Stillbirth and facility delivery output of the Upper East and Northern regions of Ghana.**

'stillbirth', born dead or dying within day zero, and '0' representing 'no stillbirth'. Also, '1' was constructed to represent perinatal death (newborn death within 6 days of life), and '0' representing 'no perinatal death'.

**Qualitative.** The two regions (Upper East and Northern regions) were zoned into three and a hospital each selected purposively for the study. Two health centers per zone were also selected to add rural context to the data. Service providers (midwives and medical doctors) from the antenatal care clinics and labor units of the selected health facilities were then selected

**Table 1. Facility deliveries versus Stillbirth in the Upper East and Northern Regions.**

| Indicator | 2016 | 2017 | 2018 | 2019 | 2020 | Total |
|---|---|---|---|---|---|---|
| **Upper East Region** | | | | | | |
| Facility Delivery | 34,053 | 34,661 | 35,917 | 38,351 | 39,211 | 182,193 |
| Fresh | 219 | 208 | 188 | 197 | 212 | 1024 |
| Macerated | 312 | 246 | 247 | 233 | 309 | 1347 |
| Stillbirth | 531 | 454 | 435 | 430 | 521 | 2371 |
| **Northern Region** | | | | | | |
| Facility Delivery | 46,645 | 52,407 | 57,056 | 61,645 | 64,092 | 281,845 |
| Fresh | 391 | 463 | 492 | 496 | 406 | 2248 |
| Macerated | 473 | 510 | 486 | 445 | 531 | 2445 |
| Stillbirth | 664 | 973 | 978 | 931 | 937 | 4693 |

**Source:** DHIMS, Ghana Health Service, 2020

purposively and conveniently i.e., doctors or midwives on duty and not busy, who consented to participate and met the selection criteria, was recruited for one-on-one in-depth interview. Pregnant women in attendance to the antenatal care clinics were also recruited from the same facilities for the focus group discussions (FGDs). The use of multiple sources of data was to explore the idea of multiple realities and this added to data verification from multiple sources as argued by Creswell [37].

## Data collection and analysis

**Tools and pretesting.** Interview guides were developed to unearth health systems challenges that confront the 'free' policy's successful operation and how that may have affected stillbirth outcomes as per our conceptual framework which was guided by Mosley and Chen's analytical framework for child survival [22, 38]. The qualitative tools were pre-tested among random midwives in a random hospital in one of the regions, which was not part of the selected study sites. All the authors then revised the tools based on our observations to include the eliciting of critical community and facility level factors relevant to our study.

**Inclusion criteria.** Only doctors and midwives with at least 3 years of working experience in the labor units and antenatal care clinics of the selected health facilities were included in the in-depth interviews, to ensure that they have adequate experience working with the 'free' maternal health care policy. Also, only pregnant women registrants of the NHIS were included in the focus group discussion.

**Exclusion criteria.** Pregnant women whose vital signs were outside the normal range using the standard of the American Psychology Association or pregnant women who were receiving treatment for a medical condition were excluded from the study. Pregnant women who were less than 16 years, considered minors under the 1992 Constitution of the Republic of Ghana were excluded from the focus group discussions.

**Sample weighting.** We generated a weighted variable by dividing v005 by 1000,000 to cater for 6 decimal places, usually not reported by DHS. We then applied the weighted variable to the GDHS survey data sets in STATA using [pw = wgt], psu (v021) strata (v022) to set the data to cater for clustering and stratification. Thereafter, all Stata command prefixes '*svy*' to take the weighting into account across the secondary data analysis. We then checked for sensitivity and overdispersion using the negative binomial regression model.

**Confounding variables.** Maternal age, area of residence, employment status, abortion history, caesarean section, marital status, and educational status were adjusted for as confounding as these were statistically significant ($p < 0.05$) with the outcome variable of interest (stillbirth and perinatal) or the exposure variable (NHIS, proxy to the 'free' maternal health care policy).

**Quality control and trustworthiness.** The sample weighting catered for clustering and stratification across rural and urban areas of the complex DHS data design. The regression analysis also used the Taylor linearization technique to achieve reduced standard error. The use of purposive sampling for the qualitative data was deliberate to achieve trustworthiness through the acquisition of information from the right sources of service providers, doctors and midwives and pregnant women as policy users. This was critical, giving the 'free' policy is implemented by medical doctors and midwives. Also, the inclusion of an expert informant, a regional director of health services validated the field data which was useful and catered for the idea of multiple sources of information [39].

**Data analysis—Quantitative.** The overall stillbirth and perinatal mortality ratios were estimated between the two rounds of DHS for comparison. We then estimated the prevalence of stillbirth and perinatal mortality between the baseline (2008) and end line (2014) to compare the outcomes, pre and post the 'free' policy intervention. Finally, we merged the two

rounds of data sets and analyzed for risk of stillbirth and perinatal mortality among the 'free' maternal health care policy using multiple logistic regression. We then tested for sensitivity using Poisson regression and checked for over-dispersion using negative binomial regression. All regression outputs are reported in Tables 2 and 3.

**Data analysis—Qualitative.** One-on-one interviews and group discussions were transcribed verbatim into Microsoft office, double-checked for accuracy, and imported into INVIVO 10 for analysis by coding the data in line with the study objectives of stillbirths and context factors of maternal health care utilization. The transcripts were read multiple times and grouped into similar and dissimilar statements with annotations. Significant statements were then categorized under constructed themes, reviewing each theme carefully for relevance.

Statements that did not align themselves to a particular theme were thoroughly examined for relevance and excluded altogether if they didn't speak to the study objective. Constructed themes were based on common phrases and similar statement approaches. Relevant statements are quoted verbatim in reporting the qualitative results to convey participants' impressions and aid explanatory power.

**Ethical consideration.** This study is part of the PhD research work of the first author and received ethical clearance from the Ghana Health Service Ethical Review Board reg. no. GHS-ERC: 002/04/19. The secondary data was obtained from Measure DHS after completing an online application process. All study participants for the primary data consented to participate in the study and completed a consent form. All interviews were conducted in private rooms, while focus group discussions were held in open spaces under chalets for aeration as part of the COVID-19 protocol. All the pregnant women had their vital signs checked by a registered nurse for normalcy prior to joining the focus group. Also, this study protocol received four double-blinded external reviewers and was published by BMC Reproductive Health Journal [22].

## Results

### Quantitative findings

**Stillbirth and perinatal death in the Upper East and Northern regions.** Antepartum stillbirth is increasing in both regions since 2019, particularly in the Upper East region of Ghana and this is reflected in the overall rise in stillbirth in the region (Fig 3) despite the increase in uptake of facility delivery (Table 1).

**Distribution of maternal and population characteristics.** As shows in Table 4, more women accessed the 'free' policy in 2014 (68%) compared to 2008 (39%). Of antenatal care uptake, 62% of the pregnant women made 4 plus visits under the free maternal health care policy group. Also, more women delivered in health facilities (65%) among the 'free' policy group, compared to the no- 'free' policy group. Of the maternal and population characteristics maternal group age (p < 0.0001), area of residence (p < 0.0001), history of abortion (p = 0.0104), employment status (p = 0.0271), maternal education (p < 0.0001), wealth index (p = 0.0001), marital status (p < 0.0001) and region (p < 0.0001) were statistically significant between the two groups.

**Prevalence of stillbirth and perinatal mortality.** In total, 174 stillbirths were recorded between 2008 and 2014 rounds of DHS. Of this, 55 (28.7%) were reported in the 2008 DHS, compared to 119 (43.1%) in 2014, showing an increase in percentage points 14.4. Also, 243 perinatal deaths were reported between 2008 and 2014 of which 88 (45.6%) were reported in 2008, compared to 155 (56.4%) in 2014, representing a 10.8 percentage point increase (Table 5).

**Table 2. Association between the free maternal health care policy and risk of stillbirth.**

| Stillbirth | Logistic regression with Linearized standard error | | | Poisson regression with linearized standard error | | | Negative binomial regression with robust std. error | | |
|---|---|---|---|---|---|---|---|---|---|
| | aOR | (CI: 95%) | P-Value | aPR | (CI: 95%) | P-value | aPR | (CI: 95%) | P-value |
| **Policy intervention** | | | | | | | | | |
| No_FMHCP | 1 | | | 1 | | | 1 | | |
| FMHCP | 1.64 | (1.02–2.65) | 0.041* | 1.34 | (1.00–1.79) | 0.045* | 1.34 | (1.00–1.79) | 0.045* |
| **Maternal age** | 1.05 | (1.01–1.08) | 0.007* | 1.02 | (1.00–1.05) | 0.006* | 1.02 | (1.00–1.05) | 0.006* |
| **Twin pregnancy** | | | | | | | | | |
| Singleton | 1 | | | 1 | | | 1 | | |
| 1st set of twins | 1.39 | (0.66–2.90) | 0.375 | 1.22 | (0.83–1.77) | 0.296 | 1.22 | (0.83–1.77) | 0.296 |
| 2nd set of twins | 1.24 | (0.43–3.56) | 0.680 | 1.15 | (0.66–2.00) | 0.609 | 1.15 | (0.66–2.00) | 0.609 |
| 3rd set of twins | - | - | - | 1.99 | (1.20–3.20) | 0.008* | 1.99 | (1.20–3.20) | 0.008* |
| **Caesarean section** | | | | | | | | | |
| No | 1 | | | 1 | | | 1 | | |
| Yes | 2.10 | (0.94–4.67) | 0.067 | 1.36 | (0.96–1.94) | 0.084 | 1.36 | (0.96–1.94) | 0.084 |
| **Abortion history** | | | | | | | | | |
| No | 1 | | | 1 | | | 1 | | |
| Yes | 1.04 | (0.59–1.84) | 0.875 | 1.02 | (0.75–1.37) | 0.906 | 1.02 | (0.75–1.37) | 0.906 |
| **Area of residence** | | | | | | | | | |
| Urban | 1 | | | 1 | | | 1 | | |
| Rural | 1.82 | (0.95–3.48) | 0.067 | 1.37 | (0.97–1.95) | 0.072 | 1.37 | (0.97–1.95) | 0.072 |
| **Educational status** | | | | | | | | | |
| No education | 1 | | | 1 | | | 1 | | |
| Primary | 1.26 | (0.66–2.40) | 0.472 | 1.14 | (0.77–1.68) | 0.505 | 1.14 | (0.77–1.68) | 0.505 |
| Secondary | 2.02 | (1.06–3.84) | 0.030* | 1.47 | (1.01–2.13) | 0.041* | 1.47 | (1.01–2.13) | 0.041* |
| Tertiary | 0.25 | (0.03–1.74) | 0.163 | 0.42 | (0.10–1.71) | 0.229 | 0.42 | (0.10–1.71) | 0.229 |
| **Wealth index** | | | | | | | | | |
| Poorest | 1 | | | 1 | | | 1 | | |
| Poorer | 0.95 | (0.48–1.88) | 0.895 | 0.99 | (0.65–1.50) | 0.966 | 0.99 | (0.65–1.50) | 0.966 |
| Middle | 1.38 | (0.65–2.90) | 0.394 | 1.22 | (0.79–1.90) | 0.362 | 1.22 | (0.79–1.90) | 0.362 |
| Richer | 1.26 | (0.87–5.84) | 0.090 | 1.52 | (0.92–2.53) | 0.101 | 1.52 | (0.92–2.53) | 0.101 |
| Richest | 1.64 | (0.53–5.04) | 0.383 | 1.33 | (0.70–2.54) | 0.378 | 1.33 | (0.70–2.54) | 0.378 |
| **Region** | | | | | | | | | |
| Western | 1 | | | 1 | | | 1 | | |
| Central | 0.74 | (0.31–1.78) | 0.508 | 0.81 | (0.48–1.36) | 0.440 | 0.81 | (0.48–1.36) | 0.440 |
| G. Accra | 1.59 | (0.50–5.04) | 0.429 | 1.29 | (0.69–2.41) | 0.407 | 1.29 | (0.69–2.41) | 0.407 |
| Volta | 2.54 | (0.74–8.64) | 0.134 | 1.55 | (0.86–2.79) | 0.141 | 1.55 | (0.86–2.79) | 0.141 |
| Eastern | 1.42 | (0.58–3.48) | 0.439 | 1.23 | (0.73–2.05) | 0.436 | 1.23 | (0.73–2.05) | 0.436 |
| Ashanti | 1.15 | (0.46–2.88) | 0.761 | 1.08 | (0.64–1.85) | 0.749 | 1.08 | (0.64–1.85) | 0.749 |
| Brong-Ahafo | 0.92 | (0.33–2.58) | 0.881 | 0.93 | (0.50–1.75) | 0.837 | 0.93 | (0.50–1.75) | 0.837 |
| Northern | 1.02 | (0.41–2.54) | 0.959 | 0.96 | (0.54–1.69) | 0.892 | 0.96 | (0.54–1.69) | 0.892 |
| Upper East | 1.08 | (0.33–3.55) | 0.888 | 1.03 | (0.47–2.22) | 0.940 | 1.03 | (0.47–2.22) | 0.940 |
| Upper West | 1.15 | (0.45–2.95) | 0.761 | 1.08 | (0.61–1.89) | 0.785 | 1.08 | (0.61–1.89) | 0.785 |

**Notation: 1** –reference; **aOR**–adjusted Odd Ratio; **aPR**–adjusted Prevalence Ratio

* $p<0.05$.

**Table 3. Association between the free maternal health care policy and risk of perinatal mortality.**

| Perinatal Mortality | Poisson regression with Linearized std. error | | | Binary logistics regression with linearized std. error | | | Negative binomial regression with linearized std. error | | |
|---|---|---|---|---|---|---|---|---|---|
| | aOR | (CI: 95%) | P-Value | aPR | (CI: 95%) | P-Value | aPR | (CI: 95%) | P-Value |
| **The Policy** | | | | | | | | | |
| No_FMHCP | 1 | | | 1 | | | 1 | | |
| FMHCP | 2.04 | (1.28–3.25) | 0.003* | 1.35 | (1.08–1.66) | 0.006* | 1.35 | (1.08–1.66) | 0.006* |
| Maternal age | 1.03 | (0.99–1.06) | 0.100 | 1.01 | (0.99–1.03) | 0.101 | 1.01 | (0.99–1.03) | 0.101 |
| **Twin pregnancy** | | | | | | | | | |
| Singleton | 1 | | | 1 | | | 1 | | |
| 1st set of twins | 1.37 | (0.67–2.80) | 0.377 | 1.14 | (0.87–1.49) | 0.331 | 1.14 | (0.87–1.49) | 0.331 |
| 2nd set of twins | 1.82 | (0.70–4.72) | 0.218 | 1.24 | (0.91–1.69) | 0.166 | 1.24 | (0.91–1.69) | 0.166 |
| 3rd set of twins | 1 | - | - | 1.43 | (1.03–1.98) | 0.033* | 1.43 | (1.03–1.98) | 0.033* |
| **Cesarean section** | | | | | | | | | |
| No | 1 | | | 1 | | | 1 | | |
| Yes | 2.02 | (0.87–4.66) | 0.100 | 1.25 | (0.97–1.60) | 0.075 | 1.25 | (0.97–1.60) | 0.075 |
| **Abortion history** | | | | | | | | | |
| No | 1 | | | 1 | | | 1 | | |
| Yes | 1.91 | (1.07–3.41) | 0.028* | 1.25 | (1.02–1.53) | 0.031* | 1.25 | (1.02–1.53) | 0.031* |
| **Area of residence** | | | | | | | | | |
| Urban | 1 | | | 1 | | | 1 | | |
| Rural | 1.28 | (0.70–2.32) | 0.407 | 1.11 | (0.87–1.42) | 0.366 | 1.11 | (0.87–1.42) | 0.366 |
| **Education** | | | | | | | | | |
| No education | 1 | | | 1 | | | 1 | | |
| Primary | 0.93 | (0.49–1.74) | 0.822 | 1.97 | (0.71–1.31) | 0.863 | 1.97 | (0.71–1.31) | 0.863 |
| Secondary | 1.89 | (0.98–3.63) | 0.056 | 1.30 | (0.97–1.72) | 0.071 | 1.30 | (0.97–1.72) | 0.071 |
| Tertiary | 0.25 | (0.04–1.44) | 0.124 | 0.52 | (0.19–1.40) | 0.196 | 0.52 | (0.19–1.40) | 0.196 |
| **Wealth index** | | | | | | | | | |
| Poorest | 1 | | | 1 | | | 1 | | |
| Poorer | 0.56 | (0.26–1.23) | 0.153 | 0.80 | (0.56–1.13) | 0.207 | 0.80 | (0.56–1.13) | 0.207 |
| Middle | 0.84 | (0.35–1.99) | 0.694 | 0.93 | (0.66–1.31) | 0.690 | 0.93 | (0.66–1.31) | 0.690 |
| Richer | 0.87 | (0.32–2.35) | 0.789 | 0.95 | (0.65–1.39) | 0.818 | 0.95 | (0.65–1.39) | 0.818 |
| Richest | 0.76 | (0.24–2.40) | 0.641 | 0.90 | (0.58–1.38) | 0.639 | 0.90 | (0.58–1.38) | 0.639 |
| **Region** | | | | | | | | | |
| Western | 1 | | | 1 | | | 1 | | |
| Central | 0.63 | (0.24–1.61) | 0.340 | 0.79 | (0.52–1.18) | 0.258 | 0.79 | (0.52–1.18) | 0.258 |
| G. Accra | 1.73 | (0.54–5.56) | 0.353 | 1.20 | (0.77–1.86) | 0.412 | 1.20 | (0.77–1.86) | 0.412 |
| Volta | 2.79 | (0.77–10.0) | 0.116 | 1.32 | (0.89–1.97) | 0.159 | 1.32 | (0.89–1.97) | 0.159 |
| Eastern | 1.36 | (0.51–3.62) | 0.526 | 1.11 | (0.76–1.62) | 0.577 | 1.11 | (0.76–1.62) | 0.577 |
| Ashanti | 1.03 | (0.38–2.78) | 0.949 | 1.00 | (0.67–1.48) | 0.986 | 1.00 | (0.67–1.48) | 0.986 |
| Brong-Ahafo | 0.87 | (0.30–2.50) | 0.809 | 0.93 | (0.60–1.43) | 0.754 | 0.93 | (0.60–1.43) | 0.754 |
| Northern | 0.59 | (0.21–1.64) | 0.313 | 0.74 | (0.47–1.17) | 0.199 | 0.74 | (0.47–1.17) | 0.199 |
| Upper East | 0.45 | (0.12–1.59) | 0.217 | 0.64 | (0.52–1.27) | 0.200 | 0.64 | (0.52–1.27) | 0.200 |
| Upper West | 0.65 | (0.23–1.85) | 0.421 | 0.81 | (0.61–1.89) | 0.375 | 0.81 | (0.61–1.89) | 0.375 |

**Notation: 1** –reference; **aOR**–adjusted Odds Ratio; **aPR**–adjusted Prevalence Ratio

*p < 0.05.

**Table 4. Distribution of maternal characteristics between the 'free' maternal health policy and the no 'free' maternal health care policy groups.**

| Description | Observation | No FMHCP (%) | FMHCP (%) | Pearson Design-based F test (p-value) |
|---|---|---|---|---|
| **DHS Year** | | | | 0.0001** |
| 2008 | 2987 | 1785 (61) | 1202 (39) | |
| 2014 | 5883 | 1796 (32) | 4087 (68) | |
| **ANC uptake** | | | | 0.0001** |
| 0–3 visits | 1021 | 607 (63) | 414 (37) | |
| 4+ visits | 5335 | 1,921 (38) | 3,414 (62) | |
| **Delivery place** | | | | 0.0001** |
| Home | 3147 | 1,705 (56) | 1,442 (44) | |
| Facility delivery | 5672 | 1,853 (35) | 3,819 (65) | |
| **Group Age** | | | | 0.0001** |
| 15–19 | 326 | 168 (53) | 161 (47) | |
| 20–24 | 1592 | 745 (49) | 847 (51) | |
| 25–29 | 2318 | 884 (40) | 1434 (60) | |
| 30–34 | 2009 | 719 (37) | 1290 (63) | |
| 35–39 | 1578 | 589 (38) | 991 (62) | |
| 40–44 | 771 | 334 (47) | 437 (53) | |
| 45–49 | 276 | 147 (59) | 129 (41) | |
| **Area of residence** | | | | 0.0001** |
| Urban | 3343 | 1125 (36) | 2218 (64) | |
| Rural | 5527 | 2456 (46) | 3071 (54) | |
| **History of Abortion** | | | | 0.0104* |
| No Abortion | 7132 | 2951 (43) | 4181 (57) | |
| Abortion | 1732 | 626 (38) | 1106 (62) | |
| **Employment status** | | | | 0.0271* |
| unemployed | 1581 | 568 (38) | 1013 (62) | |
| Employed | 7266 | 2996 (43) | 4270 (57) | |
| **Education status** | | | | 0.0001** |
| No Education | 3169 | 1412 (47) | 1757 (53) | |
| Primary Education | 1931 | 920 (52) | 1011 (48) | |
| Secondary Education | 3481 | 1196 (36) | 2285 (64) | |
| Tertiary Education | 289 | 53 (21) | 236 (79) | |
| **Wealth quintile index** | | | | 0.0001** |
| Poorest | 2854 | 1253 (47) | 1601 (53) | |
| Poorer | 1960 | 906 (50) | 1054 (50) | |
| Middle | 1587 | 63 (43) | 954 (57) | |
| Richer | 1385 | 491 (38) | 894 (62) | |
| Richest | 1084 | 298 (30) | 786 (70) | |
| **Marital status** | | | | 0.0001** |
| Never married | 534 | 261 (51) | 272 (49) | |
| Married | 6126 | 2272 (38) | 3854 (62) | |
| Divorced | 125 | 74 (66) | 51 (34) | |
| Widow | 127 | 54 (49) | 73 (51) | |
| Living together | 1681 | 781 (47) | 906 (53) | |
| Not living together | 271 | 138 (52) | 133 (48) | |

(*Continued*)

**Table 4.** (Continued)

| Description | Observation | No FMHCP (%) | FMHCP (%) | Pearson Design-based F test (p-value) |
|---|---|---|---|---|
| **Region** | | | | 0.0001** |
| Western | 852 | 359 (40) | 495 (60) | |
| Central | 830 | 467 (57) | 363 (43) | |
| Greater Accra | 739 | 337 (41) | 402 (59) | |
| Volta | 723 | 286 (40) | 437 (60) | |
| Eastern | 805 | 310 (38) | 495 (62) | |
| Ashanti | 1038 | 497 (46) | 541 (54) | |
| Brong-Ahafo | 919 | 281 (31) | 638 (69) | |
| Northern | 1381 | 607 (45) | 774 (55) | |
| Upper East | 767 | 240 (32) | 536 (68) | |
| Upper West | 807 | 199 (24) | 608 (76) | |

Significant level

* p < 0.05

** p < 0.001

**Stillbirth and perinatal mortality rate.** Stillbirth birth rate for 2008 was 19 per 1000 live births, compared to 21 per 1000 live births in 2014, while perinatal mortality rate was 31 per 1000 live births in 2008 and declined to 27 per 1000 lives birth in 2014 (Table 6).

**Risk of stillbirth in the 'free' maternal health care policy group.** Babies were 1.64 times more likely to be stillborn in the FMHCP group, compared to the no FMHCP group; aOR: 1.64; 95% CI: 1.02 to 2.65; p = 0.041. The results are similar across the Poisson and negative binomial regressions models, aPR: 1.34; 95% CI: 1.00 to 1.79; p = 0.045 respectively as shown in Table 2.

**Risk of perinatal mortality in the 'free' maternal health care policy group.** Babies were 2.04 times likely to die within 6 days of life in the FMHCP group compared to their counterparts in the no FMHCP, aOR: 2.04; 95% CI: 1.28–3.25; p = 0.003. for the results also compare similarly for the Poisson and Negative binomial regressions, aPR: 1.34 and these were statistically significant, p = 0.006, respectively. Women with a secondary level of education were more likely to register perinatal mortality compared to women with no formal education, aOR: 1.89; 95% CI: 0.98 to 3.63. However, this was not statistically significant, p = 0.056. Women with a history of abortion were also more likely to record perinatal mortality compared to women with no history of abortion, and this was statically significant, aOR: 1.91; 95% CI: 1.07 to 3.41; p = 0.028.

**Table 5. The proportion of stillbirths and perinatal mortality in 2008 and 2014.**

| Variable | Obs. | 2008 (%) | 2014 (%) | Pearson Design-based F test (p-value) |
|---|---|---|---|---|
| **Stillbirth** | | | | 0.0067* |
| Not stillborn | 313 | 143 (71.3) | 170 (56.9) | |
| Stillborn | 174 | 55 (28.7) | 119 (43.1) | |
| Total | 487 | 198 (100) | 289 (100) | |
| **Perinatal Mortality** | | | | 0.0344* |
| Not recorded | 244 | 110 (54.4) | 134 (43.6) | |
| Perinatal Death | 243 | 88 (45.6) | 155 (56.4) | |
| Total | 487 | 198 (100) | 289 (100) | |

**Table 6. Estimated stillbirth and perinatal mortality rates.**

| Variable | Rate | 2008 | 2014 |
|---|---|---|---|
| Stillbirth | Per 1000 live births | 19 | 21 |
| Perinatal Mortality | Per 1000 live births | 31 | 27 |

## Qualitative findings

**Study participants.** In all 67 service providers and pregnant women participated in the qualitative study (Table 7). Of the service providers, midwives were 18, and doctors/directors were 5. Of the pregnant women participants, 43 (98%) were married, 38 (86%) were employed, while 40 (81%) gave birth previously (Table 8).

## Common themes

*Rising stillbirths and related causes*. Stillbirth was on the rise based on the facility records and the regional data, but the service providers attributed the rise to multiple reasons, late reporting to antenatal care clinics, delayed arrival to delivery centers by pregnant women in labor and the use of local herbs as oxytocin. Service also providers argued that improved record keeping associated with increased utilization make the numbers of stillbirths look worse than it appears. In other words, pregnant women benefiting from the 'free' policy are more likely to report to and deliver in a health facility and have their data captured as compared to the previous data capture rate under out-of-pocket payment. A medical doctor and charge midwives explain.

> *"Actually, just this half–year, stillbirth numbers weren't encouraging. It was bad. We had 22, but 13 were macerated. Then we had 9 fresh stillbirths. The numbers are going up [increased]"* **(Midwife 1, IDI, Bongo District)**

> *"In the region, when you look at the picture, despite the so many interventions, one will say SBs [stillbirth] are still high. But when you look at it critically, it is the reporting which is also going up, so it makes you think that the policy is not helping"* **(Doctor 1, IDI, UER).**

> *"We have a high rate. This time we are getting mothers who are coming with Intra–Uterine Fetal Death (IUFD). This year we had 15 for the first 6 months. When you compare, I will say because we are taking records, that is why the numbers are high. . .previously there was no documentation"* **(Midwife 1, IDI, Zebilla District).**

Antepartum stillbirths were commonly recorded among facility deliveries, and service providers deemed this as a prove that babies died in utero before arrival to a health facility and blamed this on community and individual level factors rather care giver related. Another phenomenon reported was previous maternal health care history which appear to negatively influence the outcomes of stillbirth.

**Table 7. Distribution of qualitative study participant.**

| Method | Participant | Size |
|---|---|---|
| In-depth interview | Doctors/directors | 5 |
| In-depth interview | Midwives | 18 |
| Focus Group discussion | Pregnant women | 8 (44) |
| **Total** | | **67** |

**Table 8. Characteristics of pregnant women participants.**

| Description | Frequency (%) |
|---|---|
| **NHIS status** | |
| Registrant | 44 (100%) |
| Non-registrants | 0 (0%) |
| Total | 44 (100%) |
| **Marital Status** | |
| Married | 43 (98%) |
| Not married | 2 (2%) |
| Total | 44 (100%) |
| **Employment Status** | |
| Employed | 38 (86%) |
| Unemployed | 6 (14%) |
| Total | 44 (100%) |
| **Parity** | |
| Prime parity | 4 (9%) |
| Multiparty | 40 (81%) |
| Total | 44 (100%) |

Reasons for delays bothered on two items as numerated by the service providers; some pregnant women want less of vaginal examination, explaining that the experience is uncomfortable and secondly, women with previous cesarean section (CS) avoided hospitals in order not to invite another CS. Vaginal delivery is a source of pride. Pregnant women will risk delivering per vagina as a sign of womanhood. Service providers explained further as follows.

*"Few cases also dodge the hospital . . . maybe she had two previous cesarean sections (CS) and thinks that if she comes to the hospital, there will be another CS, so they avoid the hospital and when there are complications, then they quickly come. They want to deliver per vagina at all costs"* **(Doctor 1, IDI, UER)**

*". . .and when they come, we manage them at postnatal care. . .. they say they didn't know that labor had started, some say they don't want the examination. One woman was frank, she said when they come, they put fingers on her vagina and that one she doesn't like it"* **(Doctor 2, IDI, UER).**

Additionally, the cultural practice of given herbal preparations to women during labor was reported from the study sites and this seem to play a role which the midwives inferred contributed to undesirable outcome of stillbirths. Pregnant women are served locally prepared mixtures known as *kaligutiem*, to speed up uterine contraction during labor at the blind side of midwives and doctors. Essentially, the herbs act as oxytocin and potentiates the effect of prescribed medicines when combined, with lethal consequences of risk of excessive uterine contraction. Usually, mothers-in-law administers the potion before coming to the hospital or may secretly give a dose to the woman in labor if they judged a labor-process of having prolonged. Some communities are notorious for the use of the local herbs and this impedes the pathway to accessing health care during labor. The midwives shared their experiences.

*"They also take 'kaligutiem' [local oxytocin to aid uterine contraction] before coming to the hospital, and when they come the contractions will be too high"* **(Midwife 2, IDI, Sagnarigu District).**

*"Some also come with excessive contraction because of the 'kalgutiem', especially those coming from Dotoyille and Kunyevilla. The mothers–in–law. They will give it to them and follow them to hospital as well"* **(Midwife 1, IDI, Sagnarigu District).**

During the focus group discussions, pregnant women claimed knowledge of *kalgutiem* but denied usage of same, unsurprisingly.

*"We have heard about 'kuligutiem', but we don't use it. We don't know anybody who uses it. The midwives have complained and have advised us against it.* **(Pregnant Woman 3, FGD1, Sagnarigu).**

*Poor use of delivery partograph for labor monitoring.* Surprisingly, it merged that labor process and progress were poorly monitored as delivery partograph, a tool recommended by the World Health Organization was sparingly used across delivery suits in the selected hospitals. A senior medical officer and director of the regional health services noted during the expert informant interviews that,

*"Even though we use partograph to monitor labor we realize a substantial number are not monitored. Using a partograph to monitor will tell you the condition of that baby. So that if you realize that the baby has difficulties, that baby can be delivered [via caesarean section]."* **(Regional Director of Health Services, KII, UER).**

The delivery partograph, according to experts served as an indicator for initiating advance action of cesarean operation, necessary to save the life of the unborn child. Although the use of delivery partograph is not for all pregnant women, its use during the labor management process on eligible mothers is not optional, but this was not the case at the study site. The director continued,

*"Yes! A significant proportion of labor are not monitored [with delivery partograph]. Those women who are eligible, it should be 100%. . ."* **(RDHS, KII, UER).**

A senior midwife with over 10 years work experience in one of the regions also shared her experience when they carried out a monitoring exercise on behalf of UNICEF. Her observation of records of partograph use was a major concern. She claimed.

*"As for partograph dier! It is 0 out 100 in. . .hospital [a particular hospital]. We went for monitoring on behalf of UNICEF and what we saw was not good all at"* **(Midwife 2, Tamale West)**

*Little or no interest in 'matters' of stillbirth.* During the one-on-one interview sessions, a medical doctor observed that somehow, little attention is given to stillbirth issues as compared to maternal mortality. Himself as a doctor, does not get to hear about stillbirth in his ward unless there was a review of a visiting team from the regional health directorate. Not even the media were interested in stillbirths as much as they were interested in maternal mortality. The doctor added thus,

*"We don't pay attention to matters of stillbirth the way we do for maternal deaths. One mother will die and the whole hospital will hear about it. I don't even know the stillbirths in*

*the labor ward. They don't tell me. . .unless we are reporting. But when there's maternal mortality, eeeiii! even the media is interested"* **(Doctor3, IDI, UER)**

The medical doctor's view appears to align with a seemingly common practice of midwives ignoring apparent calls of pregnant women in labor, which caught the attention of one of the pregnant women and she shared her views during the group discussions.

*"Sometimes you can be crying, and they won't mind you. One time I was suffering, and the midwives didn't bother to check on me. I said my baby is coming. . ..by the time they came my baby was gone. They don't care about our babies"* **(PW4, FG3, Bongo District).**

Even though the reasons for their non-response were beyond the scope of this study, the descriptions during the focus group discussion suggest that there are underlying challenges that perhaps explains the poor monitoring of the labor process and the eventual outcomes of stillbirths in the study sites. Pregnant women participants added, thus,

*"they [midwives] don't pay attention to our babies. One woman nearly gave birth on the bench. She was calling the midwives. . .the baby is coming; the baby is coming. Oh! I felt very sad"* **(PW2, FG1, Zebilla District)**

Conversely, midwives disclosed that pregnant women had a laid-back attitude towards the survival of their unborn babies, sometimes refusing surgery as may be required and also providing inaccurate reproductive history which affects the caregiving process and hence, influencing stillbirths and perinatal deaths. Charge midwives had these to say in one of the district hospitals.

*"a woman came, and the liquor was small, so the best we could do was conduct CS. When we told them, they told us that if the water is not ok, can't you fetch water and add it. They're opposed to cesarean section. They went home. . .came back some few days later and the baby was dead. . ."* **(Midwife 2, IDI, Zebilla District)**

*"Taking history is key. . . The woman misled the midwives concerning her parity. We started inducing, and she raptured, then, we asked a relative (her daughter) and she said her mother had 6 children and 1 died. Such a person should be induced. . .we were misled."* **(Midwife 2, IDI, Bongo District)**

*Intermittent shortage of medicine commodities.* Intermittent shortages of drugs were also reported during the in-depth interviews. Pregnant women are routinely asked to purchase some medicines outside of the health facility set up to augment their required supplementary intake. Not only did this affect the economic situation of mothers and their families, but it also frustrates quality-of-care processes of the health care professionals. The medical officers shared their experiences at the antenatal clinics.

*"Our environment is not good. Personal hygiene is poor. Unlike other places where they think that labor is a sterile procedure, here, we routinely put all our clients on the antibiotic cover, whether you're on episiotomy, assisted delivery, or not."* **(Doctor 1, IDI, UER)**

*"When you visit the facility and certain medication is not available, they are written for you in a prescription. So far as our facility is a concern, if a medication is not available. . .we put it on a prescription for you to find a pharmacy shop to procure. . ."* **(Doctor 1, IDI, UER)**

*"The issue has to do with drugs. The challenge here is that most of the time the hospital runs out of stock. When they run out of stock, the patient must buy. . . because of the poverty level, most of them cannot afford the drugs. . ."* **(Doctor 2, IDI, UER)**

*"We use antibiotics and pain killers for Cesarean Section. Then we also have hematinic. The better once, usually we want them to buy those outsides. . .. eenh! And IV fluids too. There are certain times we go virtually down, they buy virtually everything"* **(Doctor 3, IDI, UER)**

Midwives also bemoaned the difficulty in getting drugs at the facility level, more so as some of these drugs are considered an emergency requirement yet not in supply, and this adversely affects the effective functions of caregiving with a direct consequence on the unborn/newborn child.

*"And after that, they pay for vitamin k, which we give to the child. . .that is when it is a normal delivery. When it is a Cesarean section, antibiotics like Cefuroxime, Amoxyclav, and Genta-mycin are ordered by the doctor. If it is not there, they go to buy. . ."* **(Midwife 2, KNNM, UER)**

*"When the dispensary does not have hematinic (iron III), you ask them to go and buy, it is a problem. . .. what about if she comes for ANC and you write for her and in the end, she goes and not buy? She will come back with anemia. . ."* **(Midwife 3, IDI, Bongo District)**

Folic acid, a dietary supplement giving during pregnancy as recommended by the World Health Organization as essential in minimizing the risk of stillbirth, is sometimes in short supply and pregnant women are told to purchase some from the open market. They shared experiences during the focus group discussion as follows.

*"Whenever we come, we have been buying the drugs. Most of the time when we come, they do write for us to go and buy the drugs. The yellow and the red drugs"* **(PW1, FG1, Bongo District)**.

*"The last time I delivered, my husband was made to buy water [intravenous fluids] for infu-sion. . .the is a drug store outside the hospital, that's where we bought it."* **(PW5, FG2, KNNM, UER)**

## Discussion

Generally, utilization outcomes improved over time between 2008 and 2014 showing statistically significant differences between the Ghana Demographic and Health Survey data. In a similar vein, there was a corresponding increase in stillbirth and perinatal mortality and although, population growth is one plausible explanation to this, the introduction of the 'free' maternal health care policy was also key to increasing utilization and this may put pressure on the health system capacity to deal newborn care, as previously reported [24, 40–42].

Stillbirth was accounted for mainly in 2014, with a statistically significant difference, $p < 0.0344$. Conversely, we found that the perinatal mortality rate declined in 2014 by 4 per 1000 live births, moving from 31 per 1000 live births in 2008 to 27 per 1000 live births in 2014. On the other hand, stillbirth rate was worse off, increasing over time by 2 per 1000 live births between 2008 and 2014.

By implication, while perinatal mortality is declining, stillbirth is rising, and this supports the views espoused by the service providers during the in-depth interviews (IDIs). The inverse

relationship between stillbirth and perinatal mortality is rather surprising because perinatal mortality feed directly on stillbirth, hence, the expectation would be that as one decreases, the other should also, but this is contrary to the current findings. Nevertheless, the current findings suggests that early neonatal deaths were declining at a factor rate, perhaps outpacing the rate of stillbirth, and hence reflecting in the overall decline in perinatal mortality rate.

Secondly, home deliveries may be recording low stillbirths, compared to facility-level deliveries. This is not farfetched given the observations during the IDIs that pregnant women in labor turn to rash to the hospital after attempting and failing home delivery. Successful home deliveries will likely be those without complications and perhaps fewer mortality outcomes, yet, the rising number of stillbirths, which is consistent with the current increase in numbers of stillbirths in sub-Saharan Africa as reported by the WHO is a worrying development and a challenge for health systems and policy [4, 7].

Both stillbirth and perinatal deaths were prevalent among the 'free' maternal health policy group compared to the no 'free' maternal health care policy group. The qualitative exploration revealed that even though the 'free' policy may have led to increase in access to maternal health care, it was against a background of shortage of medical consumables, poor and/or inadequate monitoring of pregnancy and labor process and out-of-pocket purchasing of supplementary medicines.

Iron tablets (folic acid, and ferrous sulphate) for example, are routine drugs served at antenatal clinics as supplements to prevent anemia in pregnancy and these also aid in combating stillbirth [32, 43], yet, these medicines were consistently in short supply in the selected study sites. Indeed, findings in earlier studies show that folic acid intake during pregnancy is associated with reduced stillbirth [44, 45], and therefore, intermittent shortages perhaps poses increased risk stillbirth among pregnant women. Under the current situation, quality of care is perhaps affected by the lack of medical commodities, a key component of the WHO building blocks of health systems framework [46, 47] and perhaps also affect technical quality [48].

Despite the overall reduction in the perinatal mortality rate in 2014, perinatal mortality still increased proportionately high between 2008 and 2014 by 10.8 percentage points, moving from 45.6% to 56.4% within the DHS survey period. The results compare intriguingly with the proportion of stillbirths between 2008 and 2014 which showed a much higher increase in percentage points of 14.4 in 2014 despite the introduction of the 'free' maternal health care policy.

The findings imply that stillbirth failed to show a decline in 2014 both in rate and in proportion to under 5 mortalities between 2008 and 2014, while perinatal mortality declined in overall rate but increased in proportion between 2008 and 2014. Although, this is unexpected, it represents some level of gain in the face of the 'free' policy. Arguably, the situation could have been worse without the 'free' maternal health care policy.

The prevailing health systems factors of poor monitoring, delayed arrival, inadequate attention to stillbirth 'matters' and intake of local herbs, although inconclusive probably throw some light as to why the numbers of stillbirth are high. It is imperative perhaps to consider the possibility of increased data capture and although this study did not independently explore the influence of data capture, the service providers insinuated during the IDIs that increased record keeping may have influenced the numbers of stillbirth among the 'free' maternal health care policy group.

Even though the gains in overall perinatal mortality could be due to increase in access to care at the neonatal period, and consequently improved immunization, perinatal death and stillbirth are twin concepts that work together and reasonably, a decline in one was expected to show a similar pattern in the other, unless early neonatal mortality was significantly declining [1, 30, 49].

Central to labor monitoring is the use of delivery partograph and although its implementation comes with striking challenges including form complication, midwifery staff shortages, and the lack of appreciation of its importance [50], perhaps it is imperative to state that in a situation where less partograph is used by midwives during labor, a rise in numbers of stillbirths may not be unexpected. The WHO recognized the challenges associated with using delivery partograph and approved its modification in Ethiopia, but the findings of the current study suggest yet another reason for further engagement of midwives on the need to use delivery partograph for labor monitoring.

Partograph use was more of a problem in the Upper East region than the Northern region, and although the reasons are unclear in the current study, a closer look at the regional data from DHIMS and those of the quantitative output shows that facility delivery was high in the Upper East region compared to the Northern region and therefore, increased workload may have played a role in affecting midwives' ability to use the delivery partograph.

Perceived lack of care and attention also emerged from the FGDs among the pregnant women participants. The pregnant women perception somewhat lends credence to the IDI's revelation that not much attention was paid to stillbirth 'matters' as much as maternal mortality. This is consistent with the recent report by the WHO and UNICEF, which points out that stillbirth was receiving less attention from policy and resources, and thus, it was not surprising that stillbirth declined less in the last decade of the first century compared to maternal and under 5 mortality [4, 51].

The effect of this is that pregnant women may lose trust in facility-level delivery in the long run and turn up late and in a complicated state, thus affecting care outcomes [52, 53]. Stillbirth and perinatal mortality are sensitive indicators of health systems' weaknesses and a test of the quality of care dimensions [54].

The use of *kaligutiem* to speed up contraction in labor, unaware of its adverse effect on the unborn child also emerged from the IDIs. It seems a given that women in labor will want a fast-track process of labor, yet the practice of using local herbs against medical advice demonstrates some lack of confidence in the modern health care system. The account of midwives suggests that pregnant women who take the are at risk of excessive uterine contraction with increased risk of uterine rupture and therefore stillbirth.

## Strength and limitation

The use of DHS data sets was appropriate to achieve external validity and generalizability as the data was large enough and representative. Suffice to say, the 'free' policy primary intent was to increase utilization and access to maternal health care. This study measured stillbirth and perinatal mortality outcomes in Ghana relative to the 'free' maternal health care policy using a mixed method design and this added context to the study findings and discourse.

The study show limitation as well. The quantitative analysis was based on association using regression models. Although the analysis used multivariate regression models to test sensitivity, a quasi-experimental design would perhaps have measured the treatment impact with precision and give robust results. On the qualitative side, the selected regions were from the northern section Ghana, based on the quantitative findings of stillbirth outcomes, thus, excluding pregnant women's perspective of the 'free' policy from the southern section of Ghana.

## Conclusion

Although perinatal mortality rate declined overall in 2014, stillbirth rate increased within the period suggesting a significant decline in neonatal mortality. This is a gain in that, while the

'free' maternal health care policy is yet to translate to reduced stillbirths, early neonatal mortality is declining. Giving these findings are within the health systems context of poor monitoring of labor process, and intermittent shortage of drug consumables for pregnant women, the factors may have exerted negative influence on the outcomes of stillbirths. Delayed arrival during labor and the intake of local oxytocin may be compounding pregnancy outcomes and consequently, increasing stillbirth. It is recommended that health system thinking approach be adopted by the MoH and GHS to ensure regular supply of drug supplements and outright stoppage of local herb usage among pregnant women for better outcomes. There is the urgent need for leadership to monitor the use of the delivery partograph in managing eligible mother in labor.

## Supporting information

**S1 File.**
(DOCX)

## Acknowledgments

The DHS Programme funded by the USAID is hereby acknowledged for their role and ownership of the DHS data sets. We also acknowledge Ghana Health Service and DHIMS for allowing us access data for aspect of this study. We appreciate the cooperation of the directors who granted entrée to hospitals and health centres for the primary data. Finally, we acknowledge the voluntary participants who volunteered to participate in the study.

## Author Contributions

**Conceptualization:** John Azaare.

**Data curation:** John Azaare.

**Formal analysis:** John Azaare, Patricia Akweongo, Genevieve Cecilia Aryeteey, Duah Dwomoh.

**Investigation:** John Azaare.

**Methodology:** John Azaare, Patricia Akweongo, Genevieve Cecilia Aryeteey, Duah Dwomoh.

**Project administration:** John Azaare.

**Software:** John Azaare, Duah Dwomoh.

**Supervision:** Patricia Akweongo, Genevieve Cecilia Aryeteey, Duah Dwomoh.

**Validation:** Patricia Akweongo, Genevieve Cecilia Aryeteey, Duah Dwomoh.

**Writing – original draft:** John Azaare.

**Writing – review & editing:** John Azaare, Patricia Akweongo, Genevieve Cecilia Aryeteey, Duah Dwomoh.

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
