## [Editor Report · Decision Letter 0]

29 Nov 2021

PONE-D-21-35524Evaluating the impact of maternal health care policy on stillbirth and perinatal mortality in Ghana: a mix method approach using two rounds of Ghana demographic and health survey data sets and qualitative design techniquePLOS ONE

Dear Dr. Azaare,

Thank you for submitting your manuscript to PLOS ONE. After careful consideration, we feel that it has merit but does not fully meet PLOS ONE’s publication criteria as it currently stands. Therefore, we invite you to submit a revised version of the manuscript that addresses the points raised during the review process.

We look forward to receiving your revised manuscript.

Kind regards,

Angela Lupattelli, PhD

Academic Editor

PLOS ONE

Journal Requirements:

(We also acknowledge the cooperation and support of Ghana Health Service and the Directors who granted entrée to hospitals and health centers for this study.)

3. We noted in your submission details that a portion of your manuscript may have been presented or published elsewhere. Please clarify whether this publication was peer-reviewed and formally published. If this work was previously peer-reviewed and published, in the cover letter please provide the reason that this work does not constitute dual publication and should be included in the current manuscript.

5. We note that Figure 2 in your submission contain [map/satellite] images which may be copyrighted. All PLOS content is published under the Creative Commons Attribution License (CC BY 4.0), which means that the manuscript, images, and Supporting Information files will be freely available online, and any third party is permitted to access, download, copy, distribute, and use these materials in any way, even commercially, with proper attribution. For these reasons, we cannot publish previously copyrighted maps or satellite images created using proprietary data, such as Google software (Google Maps, Street View, and Earth). For more information, see our copyright guidelines: http://journals.plos.org/plosone/s/licenses-and-copyright.

A. You may seek permission from the original copyright holder of Figure 2 to publish the content specifically under the CC BY 4.0 license.  

 In the figure caption of the copyrighted figure, please include the following text: “Reprinted from [ref] under a CC BY license, with permission from [name of publisher], original copyright [original copyright year].

B. If you are unable to obtain permission from the original copyright holder to publish these figures under the CC BY 4.0 license or if the copyright holder’s requirements are incompatible with the CC BY 4.0 license, please either i) remove the figure or ii) supply a replacement figure that complies with the CC BY 4.0 license. Please check copyright information on all replacement figures and update the figure caption with source information. If applicable, please specify in the figure caption text when a figure is similar but not identical to the original image and is therefore for illustrative purposes only.

6. We note you have included a table to which you do not refer in the text of your manuscript. Please ensure that you refer to Table 7 in your text; if accepted, production will need this reference to link the reader to the Table.

Additional Editor Comments:

Dear author,

I do have some request regarding the structure of your article, before it can be assessed further.

1) Please format the article in a way that can be more easily assessed. The Introduction is extensively long; please shorten it and make it to the point. The "contextual definition" does not need to have a specific subsection; these are terms that you can define in the text.

2) Please state your objectives clearly.

3) Please remove the formulas for generating ratios, that is superfluous.

4) Please summarize the amount of results presented in the article, in terms of number of tables, figures and text - consider moving not essential data to the supplementary material if possible.

---

## [Author Response · Author response to Decision Letter 0]

24 Feb 2022

The Academic Editor

PLOS ONE Journal 

Dear Editor,

RE: RESPONSE TO REVIEWERS: PONE-D-21-35524 - EVALUATING THE IMPACT OF MATERNAL HEALTH CARE POLICY ON STILLBIRTH AND PERINATAL MORTALITY IN GHANA: A MIX METHOD APPROACH USING TWO ROUNDS OF GHANA DEMOGRAPHIC AND HEALTH SURVEY DATA SETS AND QUALITATIVE DESIGN TECHNIQUE

1. The manuscript formatting has been revised per PLOS journal requirements including Tables and Figures. We are willing to revise further if necessary, any aspect that does not meet the requirement of the journal. However, this may have to be pointed out clearly to us to guide us.

2. The study objective has been restated to make it much clearer than before.

3. The size of the manuscript has also been reduced particularly in the introduction and discussion sections:

a. the contextual definition sub-section has been taken out and key definition incorporated into the introduction. 

b. the formula for the calculation of stillbirth and perinatal mortality ratios has also been removed from the manuscript.

c. the Tables has been reduced and others merged as necessary. We now have 5 Tables instead of 7 as contained in the original submission. 

4. You noted that some aspects of this manuscript appear to have been published somewhere else. This is not exactly clear to us. However; 

a. the original protocol design of the study was published by BMC reproductive health journal and assigned DOI (https://doi.org/10.1186/s12978-020-01011-9) which was just the methodology of the full protocol at its design stage and did not include any results. The conceptual framework in the protocol appears similar to the one in the current manuscript, but not the same. There has been a significant modification as the study progressed which is what has been submitted in the current manuscript. 

b. In preparing this manuscript, aspects of the qualitative results were presented by research gate as a pre-publication text (not peer-reviewed), usually meant to gather useful comments to enrich the manuscript (and does not constitute a publication of our current results) and thus, the wording may be similar. Accordingly, we can confirm that no part of this manuscript results has been published in any journal as a research finding and should be considered for publication in PLOS Journal as an original piece of work.

5. (We also acknowledge the cooperation and support of Ghana Health Service and the Directors who granted entrée to hospitals and health centres for this study.)

6. The above statement (item 5) was quoted to us, and intimating that we may have received funding support from Ghana Health Service or another agency. Perhaps, we did not convey the appropriate impression in the choice of words and terms such as “support of Ghana Health Service…”. However, for the avoidance of doubt, no funding support was received from Ghana Health Service or any agency of local or international in nature regarding this study. This manuscript is part of the PhD research work of the first author during his candidature in the University of Ghana School of Public Health, where the 2nd, 3rd and 4th authors are faculty and supervising committee members of the 1st author. The first author did receive tuition fee support (not research) from the Ghana Education Trust Fund (GetFUND) and this has been duly acknowledged in the manuscript. We are willing to modify our statement if the journal finds the GetFUND tuition fee as funding support for the research work.

7. Figure 2 (Fig. 2) attached in the manuscript is an original map professionally constructed for this study and about the selected study site from two regions of Ghana. The map is an original work designed for it purpose following the commencement of the study and has neither been used by anyone nor published anywhere before. To this end, we do not find the need to follow your guideline on copyright concerning third party use of a figure (the map), as this does not apply to us in this case. We are willing to remove the particular figure (Fig. 2) if there is evidence to the contrary.

8. The aspect of the ethical statement has been revised accordingly in the method section of the manuscript. The full name of the ERB, Ghana Health Service Ethical Review Board has been stated under the sub-section ‘Ethical consideration’.

9. All Tables and Figures have been duly referred to i.e. Table 7 which was cited as not referred to has been corrected. The particular Table was referenced as multiple Tables in the manuscript. However, in the revised version, Table 7 (now Table 6) is attached as a supplementary file.

10. The figures (Fig 1 and 2) have been uploaded to the Preflight Analysis and Conversion Engine (PACE) digital diagnostic accordingly to meet PLOS requirements.

11. Humbly submitted.

---

## [Decision Letter · Decision Letter 1]

16 May 2022

PONE-D-21-35524R1Evaluating the impact of maternal health care policy on stillbirth and perinatal mortality in Ghana: a mix method approach using two rounds of Ghana demographic and health survey data sets and qualitative design techniquePLOS ONE

Dear Dr. Azaare,

Thank you for submitting your manuscript to PLOS ONE. After careful consideration, we feel that it has merit but does not fully meet PLOS ONE’s publication criteria as it currently stands. Therefore, we invite you to submit a revised version of the manuscript that addresses the points raised during the review process.

We look forward to receiving your revised manuscript.

Kind regards,

Angela Lupattelli, PhD

Academic Editor

PLOS ONE

Reviewers' comments:

Reviewer's Responses to Questions

2. Is the manuscript technically sound, and do the data support the conclusions?

Reviewer #1: Yes

Reviewer #2: Yes

3. Has the statistical analysis been performed appropriately and rigorously? 

Reviewer #1: Yes

Reviewer #2: Yes

4. Have the authors made all data underlying the findings in their manuscript fully available?

Reviewer #1: No

Reviewer #2: Yes

5. Is the manuscript presented in an intelligible fashion and written in standard English?

Reviewer #1: Yes

Reviewer #2: No

6. Review Comments to the Author

Reviewer #1: Thank you for a good work. Please see manuscript text for comments. The purpose of the study has to be rewritten and stated clearly as per the title of study.

Reviewer #2: the manuscripts makes an important contribution to the field but needs major revisions. the writing style is not very much scientific> the authors would benefit much by reading latest publication in the area and adjust the manuscript accordingly. the discussion and conclusion section in their current state need further revisions as they do not meet the minimum journal standards

7. PLOS authors have the option to publish the peer review history of their article (what does this mean?). If published, this will include your full peer review and any attached files.

Reviewer #1: **Yes: **Emmanuel Ugwa

Reviewer #2: No

---

## [Author Response · Author response to Decision Letter 1]

29 Jun 2022

Dear EDITOR,

RESPONSE TO REVIEWERS COMMENTS

Manuscript Title: Evaluating the impact of maternal health care policy on stillbirth and perinatal mortality in Ghana: a mix method approach using two rounds of Ghana demographic and health survey data sets and qualitative design technique

Abstract

All comments under the ABSTRACT sub-section have been addressed as follows;

1. Background: This has been revised appropriately and the WHO/UNICEF, 2020 report referenced.

2. Methods: The quantitative data was analyzed first, then the findings necessitated the qualitative study. The IDIs, especially the expert views were born out of the findings of the quantitative findings. The interview guide is attached as supplementary file.

3. Results: Although the results of the qualitative findings provide a certain perspective to the quantitative finings, this study does not seek to establish causal inferences between the results of the quantitative analysis and the qualitative interviews. Its aim was to seek understanding within what contest the ‘free’ policy was expected to produce the desire impact.

4. Conclusion: The qualitative results did not report exactly what was or what was not part of the ‘free’ maternal health care policy. It reported on what context factors affected the ‘free’ policy implementation and therefore, a consequence the stillbirth outcomes, despite the ‘free’ policy. This has been clarified in the introduction and mothed section.

5. Key works; “policy evaluation” been included accordingly.

Introduction

1. Comment: Stillbirth and perinatal mortality are part of the maternal and child health indicators which are very sensitive health system indicators reflecting negatively on the overall performance of the health systems. This needs to come out clearly.

Response: Sentence has accordingly been revised. 

2. Comment: Stillbirth and perinatal mortality have received relatively good policy attention. E.g. new born health has received renewed attention and it has the potential of addressing perinatal mortality. Stillbirth too has received relative attention. The ENAP set targets for national governments to adopt towards addressing the two. You should reflect on what Ghana has done in line to achieve the ENAP targets.

Response: Although not explicitly stated, the ‘free’ maternal health care is implicitly implemented to achieve ascertain level of access to antenatal care and facility delivery utilization, which essentially feed into the purpose of ENAP. This has been explained in the opening introduction leading to stating the purpose of the study.

3. Comment: The global burden of 2.6 stillbirth is an old statistic. The current burden is 2 million annual stillbirths according to UNICEF/WHO report titled “The neglected tragedy; the global burden of stillbirth 2020. The authors need to use Up to date references.

Response: This has been revised.

4. Comment: Technically, the WHO defines stillbirth as intrauterine death after conception”. This statement is not true. The definition only applies to fetal death after 28 weeks. Revise as appropriate.

Response: Comments revised as suggested.

5. Comment: “Access to maternal health care is proven to reduce stillbirth”. It is not access to any maternal healthcare but rather “quality and timely maternal healthcare” you need to revise this to reflect the recommendation.

Response: “Quality and timely access” added to said sentence as recommended.

6. Comment: Related to the above, the authors need to highlight the global burden, regional picture and country status to bring out the problem clearly in a funnel-like approach.

Response: revised.

7. Comment: Did that policy explicitly say that it targeted to address stillbirth and perinatal mortality? You need to reflect it in the introduction.

Response: The policy included comprehensive antenatal care, facility level delivery and post-delivery care up to 10 days. During this period care, one of the gains expected by implication is reduced stillbirths and perinatal death. This is the bases for this study, to evaluate the ‘free’ policy contribution to late pregnancy outcomes and new-born care.

8. Comment: Policy components of the free maternal healthcare policy need to be highlighted for the reader. Right now it is not known.

Response: This has been revised in the introductory section.

9. Comment: Other than the social health insurance, the main maternal and child health intervention since 2000 has been the MDGs with specific goals for maternal and child health whose intervention impacts directly on the indicators you are studying in this study. This relationship needs to come out explicitly. 

Response: Since MDGs were reviewed to SDGs, this is perhaps, a more appropriate reference to make which is what we made in our introduction section. 

10. Comment: As early stated, other than quoting the 2015 statistics, it’s better to use recent data from the 2020 UNICEF/WHO report.

Response: The UNICEF/WHO reported has been used to update the figures and appropriated referenced. 

11. Comment: Also, instead of quoting Nigerian data it would be important to quote the stillbirth burden in Ghana straight away.

Response: Ghana data used instead of Nigeria. 

12. Comment: The definition of stillbirth is not clear; key issues to note are the 28 weeks’ threshold as the official position of WHO, and birth with no sign of life either fresh or macerated. This needs to come out explicitly and when using the official definition of WHO, it’s better to use their own sources.

Response: Statement revised.

13. Comment: Other definitions which stretch to less than 28 weeks of gestation are not official but rather country and study specific definitions based on period of viability.

Response: Definitions has been edited and revised. 

14. Comment: There are contextual factors which need to come out clearly in the background. For example, the MDGs were a cornerstone for improvement of maternal and child health services globally. Also, the UN secretary general strategy Every Woman Every Child which was operationalized through every New Born Action Plan (ENAP) need to be highlighted when investigating stillbirth. They are the back born for the renewed interest in global stillbirth campaigns.

Response: Comments appropriately incorporated.

15. Comment: An explanation of the “free maternal health policy within the national health insurance scheme needs to be done in the introduction. Was it only aimed at improving access through addressing the healthcare costs? Or there are maternal health interventions within such as increased access to EmOC, MPDSR among others.

Response: The ‘free’ maternal health care policy explained and the national health insurance explained. 

16. Comment: “Studies have shown that maternal age, rural/urban area of residence”. This paragraph ignores the important contribution of the health systems factors in addressing stillbirth risks and yet the said policy was implemented within the health systems. It would be important to strike a balance by highlighting the health systems related factors too.

Response: indeed, the policy is being implemented with certain health systems factors, which was the focus of the qualitative study. Accordingly, the sentence has been revised.

17. Comment: Contextual definitions: Stillbirth; kindly check this definition. It lacks the basics of what would constitute a stillbirth such as after the 28 weeks cut off.

Response: Definition revised. 

18. Comment: Free maternal healthcare policy: This is simplistic. I suggest an elaboration of the ingredients of this policy and the aspects of maternal health that were covered. Its coverage within the Ghanaian population and any other important information.

Response: This has been revised and scope and package included in the introduction section. 

19. Comment: Page 13 “local contest” did you want to mean “context”? if so change as appropriate.

Response: Yes, this refers to ‘context. Appropriately edited. 

20. Comments: “we analyze the outcomes of stillbirth and perinatal deaths among mothers in2008, when the policy had just started, compared to 2014 when the policy was fully rolled out”. This should speak directly to the title of the paper and objective of the study in the introduction of your abstract.

Response: Comments revised accordingly.

Methods

1. Comment: Study design; the section is rather speaking more to the study setting other than the design. Be explicit and intentioned right away.

Response: This comment has been addressed.

2. Comment: Study setting; some information given is irrelevant to this paper, rather give information that adds value. For example, the geographical location may not be that needed, instead data on stillbirth and perinatal mortality may be more important here. 

Response: Data on stillbirth for the qualitative study sites have been included.

3. Comment: The sampling procedure for the qualitative component is missing.

Response: Perhaps, this was clear enough. It has been revised accordingly.

4. Comment: Details about the qualitative tool are missing. It would be important to take the reader through the kind of information the tool elicited.

Response: A little more detail has been added on the qualitative information. Tools also attached as supplementary file.

5. Comment: Instead of attaching the ethics review letter, better quote the reference number for the protocol from the ethics committee.

Response: protocol number quoted under the section ‘ethical consideration’

6. Comment: No mention of the inclusion and exclusion criteria for FGD participants. It needs to come out.

Response: This had been revised to include inclusion and exclusion criteria of FDGs. 

7. Comments: On the qualitative data analysis, how was the codebook developed, who coded the data? How many people were involved in the analysis?

Response: The first author analysed the qualitative data and supervised by the second and third authors. Coding essentially was guided by the deductive themes of stillbirth, and the context factors affecting maternal health care utilization under the ‘free’ policy and how they affect ‘stillbirth’ outcome. The emerged significant statements were grouped as inductive sub-themes and reported as the study results with verbatim quotes. Statements were reviewed thoroughly for significance through multiple reading and grouped as affecting quality of care and perhaps negatively influencing stillbirth.

8. Comment: How was saturation attained?

Response: When subsequent IDIs added no particular new information, we determined that saturation was reached, and suspended interviews. However, FGDs continued as planned and all 8 groups of FGDs were carried out and recorded views transcribed analysed in unison. 

9. Comment: Attribution of the reduction to the policy

Response: This comment is not particularly clear. However, the study found that perinatal mortality rate declined between 2008 and 2014, while stillbirth rate increased within the same period. Yet, both perinatal mortality and stillbirth were more likely to occur in the ‘free’ maternal health care policy group compared to the no ‘free’ maternal health care policy group.

10. Comments: The authors need to clarify explicitly how the two methods of data collection were integrated and informed each other. They should also state if this did not happen and why.

Response: data collection and integration has been revised under the methods section

11. Comment: I would suggest that you follow the COREQ Checklist to report on your qualitative findings better.

Response: Although, the Consolidated Criteria for Reporting Qualitative research (COREQ) was not particularly used in the study, nonetheless, aspects in the manuscript have addressed issues the requirements of the COREQ checklist. The qualitative study conducted mainly by the first author, as a PhD candidate of the University of Ghana School of Public Health, under a thesis supervisory committee led by the second author, received ethical approval from the Ghana Health Service Ethical Review Committee (number quoted in main test of manuscript). Analysis was content in nature, where sub-themes were constructed to report the study results. 

Results 

1. Comment: “On risk of stillbirth, pregnancies in the ‘Free’ Maternal Health Policy (FMHCP) group were 1.64 times more likely to result in stillbirth compared to the no FMHCP group”. This needs to come out explicitly in the methods section how these groups were created.

Response: Group creation has been revised accordingly in the design section. Also, Table 2 has been revised to report the values of the two groups.

2. Comment: Qualitative study participants: the unit of analysis for the FGDs is the group. Therefore, when describing your study participants, its better to refer to how many FGDs were conducted other than the total number of participants within those FGDs.

Response: descriptive section of FDGs revised appropriately.

3. Comment: “The majority of pregnant women respondents had given birth previously, 41 (90.1%). Only 4 (8.9%) pregnant women participants were having children for the first time”. To me these appear to belong to the same group unless if you want to say that some were pregnant at the time of the interview and reference is being made to the index pregnancy. Revise statement to bring out clarity.

Response: Yes, we agree with your comment. The section/sentence has be revised. 

4. Comment: Table 6: the column on average duration is less significant. This description should be in the narrative and refer to the average duration of each interview. In its current form it appears to be the total duration of all interviews in each category. Kindly revise as appropriate.

Response: We agree with the reviewer. The particular column has be deleted appropriately.

5. Comment: “…A few cases also dodge the hospital may they have two previous CS and knows that if they come to the hospital, there will be CS, so they avoid the hospital, when there are complications, then they quickly come...” (Doctor 1, IDI, UER). Start to use “signposts” in your write up. Whereas the introductory statement was speaking to increased number of macerated stillbirth, this statement seem to be off. Revise as appropriate.

Response: Statement/section revised appropriately.

6. Comment: “…and when they come we manage them at postnatal care…. they say they didn’t know that labour had started, some say they don’t want the examination. One woman was frank, she said when they come, they put fingers on her vagina and that one she doesn’t like it…” (Doctor 2, IDI, UER). Refer to comment above for this quotation.

Response: Also revised.

7. Comment: Service providers also observed a culture of little or no attention to stillbirth issues. Public or media lack of attention on stillbirth was cited as a principal reason compared to maternal mortality. On the other hand, pregnant women midwives’ snobbish attitude to them during labour, as a contributory factor to the rising stillbirths. The following quotes explains further. Was this part of your study objective or scope of the study? To me it appears not.

Response: Yes, these emerged from the IDIs, as some of the contextual factors that bothers on stillbirth relative the ‘free’ policy in practice. 

8. Comment: “Use of local herbs for ‘rapid’ uterine contraction” this can’t be a subtheme of the study since its not part of the policy. It is rather inadequate implementation of the policy that is resulting into this. The write up should reflect that. Its either poor access to available services or poor enforcement of the policy that is resulting into this

Response: This has been revised and reported under ‘stillbirth’. Essentially, the IDIs explained the consequences of taking local herbs which was a common practice in some of the study sites. Thus, making it a relevant observation, as a factor associated with the poor outcome of stillbirth, despite the ‘free’ policy.

9. Comment: The qualitative findings only talk about the high numbers of stillbirth which have been explained to be caused by poor use of partograph for pregnancy monitoring and shortage of medicines. I highly doubt this is comprehensive and can solely explain the quantitative findings. More exploration of the qualitative data for more reasons to explain the high numbers.

Response: Indeed, poor use of partograph as found in the current study is one of the context factor which may be explaining the stillbirth outcomes in the study site. Other context factors that emerged included; late reporting/arrival, poor, shortages of folic acid/multivitamin, and lack of interest in stillbirth ‘matter’. The section has accordingly been revised. 

10. Comment: “Service providers are of the view that the increase in utilization puts pressure”. Ensure to write the results section in past tense since activities and comments happened sometime back.

Response: The results section has been read and revised. 

11. Comment: On the other hand, pregnant women midwives’ snobbish attitude to them during labour”. Revise sentence as appropriate.

Response: sentence has been revised.

12. Comment: Just so to confirm are these FGD quotes from mothers that experienced a stillbirth? If so it should be reported as such and if not you need to clarify in the methodology section.

Response: Not only mothers who experienced stillbirth, but mothers previous experience while giving birth at the facility level and use of using antenatal care.

13. Comment: Due to the facility level shortages, pregnant women regularly asked to purchased medicines outside of the health facility”. Revise statement as appropriate, it is not clear in its current form.

Response: Statement revised accordingly.

14. Comment: “This was not only a problem to the pregnant women, but also frustrates the quality of processes of the health care professionals” use past tense as appropriate.

Response: Manuscript read and edited accordingly.

Discussion

1. Comments: Revise the discussion section accordingly after effecting changes in the results section

Response: Discussion section revised.

2. Comment: “improvement in perinatal mortality” is it improvement in perinatal mortality or reduction in perinatal mortality?

Response: reduction in perinatal mortality. This has been revised accordingly.

Conclusion

1. Comment: Revise the discussion section accordingly after effecting changes in the results section

Response: Relevant revisions effected, including the conclusion section.

John Azaare (PhD Candidate, Thesis defended successfully)

Lead/Corresponding Author

---

## [Decision Letter · Decision Letter 2]

21 Jul 2022

PONE-D-21-35524R2Evaluating the impact of maternal health care policy on stillbirth and perinatal mortality in Ghana: a mix method approach using two rounds of Ghana demographic and health survey data sets and qualitative design techniquePLOS ONE

Dear Dr. Azaare,

Thank you for submitting your manuscript to PLOS ONE. After careful consideration, we feel that it has merit but does not fully meet PLOS ONE’s publication criteria as it currently stands. Therefore, we invite you to submit a revised version of the manuscript that addresses the points raised during the review process.

We look forward to receiving your revised manuscript.

Kind regards,

Angela Lupattelli, PhD

Academic Editor

PLOS ONE

Journal Requirements:

Reviewers' comments:

Reviewer's Responses to Questions

**Comments to the Author**

1. If the authors have adequately addressed your comments raised in a previous round of review and you feel that this manuscript is now acceptable for publication, you may indicate that here to bypass the “Comments to the Author” section, enter your conflict of interest statement in the “Confidential to Editor” section, and submit your "Accept" recommendation.

Reviewer #1: All comments have been addressed

Reviewer #2: (No Response)

2. Is the manuscript technically sound, and do the data support the conclusions?

Reviewer #1: Yes

Reviewer #2: Yes

3. Has the statistical analysis been performed appropriately and rigorously? 

Reviewer #1: Yes

Reviewer #2: Yes

4. Have the authors made all data underlying the findings in their manuscript fully available?

Reviewer #1: Yes

Reviewer #2: Yes

5. Is the manuscript presented in an intelligible fashion and written in standard English?

Reviewer #1: Yes

Reviewer #2: Yes

6. Review Comments to the Author

Reviewer #1: Thank you for the attempted revision. However, the reviewer would like to read changes made. You can either copy and paste on your comments or provide page and line numbers so that reviewer can easily identify such changes. Comments such as ''revision has been made as appropriate'' are not required.

Reviewer #2: Second round of review and the authors have done a good job to address earlier comments. more additional comments to enhance clarity are herewith attached.

regards

7. PLOS authors have the option to publish the peer review history of their article (what does this mean?). If published, this will include your full peer review and any attached files.

Reviewer #1: **Yes: **Emmanuel Ugwa

Reviewer #2: No

---

## [Author Response · Author response to Decision Letter 2]

25 Aug 2022

THE ACADEMIC EDITOR

PLOS ONE JOURNAL

Dear Editor, 

RE: Evaluating the impact of maternal health care policy on stillbirth and perinatal mortality in Ghana: a mix method approach using two rounds of Ghana demographic and health survey data sets and qualitative design technique.

Thank you for reviewing this manuscript. Below are the responses to the issues raised.

General comments.

1. The definition of stillbirth should come upfront within the first paragraph of the introductory section. Referring to it as a neglected tragedy is not a definition parse but rather what it is.

Response: This has been addressed in Page 1, paragraph 1.

2. Paragraphs two and three are a repetition with one showing absolute numbers and the other showing percentages of global stillbirths. Consider deleting one.

Response: Paragraph three has been deleted, while paragraph two revised.

3. The definition of stillbirth is inadequate and its contextual. Rather use the WHO definition which restricts stillbirth to loss after 28 weeks for global comparisons. It is clear that in developed countries the definition can go as low as 20 weeks due to viability but that is not the standard definition.

Response: This has been revised in page 4, (under ‘Context definition section).

4. It may not necessarily be history of abortion “Studies have shown that maternal age, rural/urban area of residence, twin pregnancy, history of abortion” but rather negative pregnancy history. It may be pre-term delivery, miscourage, abortion or even stillbirth. You may consider revising the sentence to bring out this fact.

Response: “history of abortion” revised appropriately in paragraph four (under ‘conceptual framework’ section).

5. Under contextual definitions, you definition of stillbirth is not appropriate. The standard definition is well known and if any adjustments to the definitions were done in this study, kindly indicate so. Key to the definition is the cut off of 28 weeks and being born with no sign of life. You may need to revise or operationalise what you call “dying within day zero after birth” for which I would highly discourage you from using.

Response: This has been revised in page 6 (under context definition).

6. Study design: Are you sure the mixed methods design was in “two prongs”? To me it appears it was sequential mixed methods because you started off with the quantitative secondary data and later the qualitative. If that is the case, then revise as appropriate.

Response: This has been revised in paragraph one, page 8 (under the ‘design section’). 

7. Statement not clear “who were not necessarily stillborn babies”. Did you want to say the mothers who participated in the FGDs were not necessarily “mothers” to stillborn babies? If that is the case, then revise as appropriate.

Response: This has been revised in paragraph two, page 8 (under the ‘design section’).

8. Qualitative study setting: other than giving a geographical description, I would rather you write more about the health systems characteristics especially as they relate to maternal health service access and stillbirth burden, or characteristics like to lead mothers to having a stillbirth. Still more about the policy under review would work more compared to giving geographical descriptions. The map can be retained though for visual clarity about the study location.

Response: The two regions are similar in structure and organization relative to health system characteristics, albeit with different outcomes of stillbirth and facility utilization. This has been reported in the manuscript (Table 1) and captured as Figure 3 (page 15).

9. Inclusion and exclusion criteria: in the inclusion you say you included women 15-49 years and in exclusion you indicate women below 16 years. Where does that put the ones aged 15 years you indicated under inclusion criteria? Were they included or excluded?

Response: Age “15-49’ as captured in the inclusion criteria is in reference to the age range of the secondary data sets as contained in the DHS survey data sets. However, age 16 years and below were excluded from the in-depth interview in line with the legal age limit for females to have consensual sexual decision, thus, pregnancy. To avoid any confusion for readers, the age range of ‘15-49’ in respect of the Ghana DHS survey data set has been removed under the inclusion criteria section.

10. Besides exclusion is not the direct opposite of inclusion criteria but that those respondents who met the inclusion criteria in the first place and for some reason were excluded from the sample later such as those that met the inclusion requirements but had missing data for other variables. Revise as appropriate.

Response: This has been revised under the section ‘Inclusion criteria’. 

11. Data analysis-qualitative: how was the code book developed? who coded the data? how was quality control ensured? how did you choose the quote to represent others in the results section?

Response: The first author coded the data and was supervised by the second, and third author. Codes were deductive (prior to conducting in-depth interviews) based on the study objectives of stillbirth and context factors of service utilization. The second and third author checked for quality of codes, clarity and uniqueness, as supervisors of the first author. Afterwards, common themes were constructed inductively through multiple reading of the transcripts. This has been explained in paragraph on of the ‘qualitative analysis’ section.

12. Ethics statements are repeated under “tools and pre-testing” and “ethical consideration”. Consider removing one to avoid repeating yourself.

Response: The ‘ethical statement’ under the “tools and pre-testing” has been removed. 

13. Results: source of data presented in table 1, how is that data from the field for you to label it as “field data”. It is clearly from the DHIMS and should be labelled as such.

Response: This has been corrected in Table (page 15).

14. Quantitative findings. Is data from table 1 not part of the quantitative findings? If so then the sub heading of quantitative findings should be shifted up to include results from table one.

Response: This comment has been addressed under the ‘Results’ section in page 14.

15. Rates of stillbirth increasing between the two timepoints this is a unique finding that needs to be explored more. Why the risk appeared to be more in the free maternal health care policy group also needs to be explored.

Response: We agree with the reviewer. The qualitative aspect of the study was aimed to address this concern, thus, the results as presented in the qualitative section. Of course, further exploration will be useful perhaps in future studies.

16. Although there was an overall decline between the two timepoints, the risk of perinatal mortality was also high in the free maternal healthcare policy compared to the control group. This phenomenon needs to be explored and discussed in relation to the national context. It calls for interrogation of the quality of secondary data used. Otherwise the conclusion would be that the free maternal healthcare policy instead led to an increase in both stillbirth and perinatal mortality.

Response: The secondary data used were two-rounds of Ghana DHS data sets (baseline and end line). DHS is a standardly collected data sets using complex design by the DHS Programme, funded by the USAID (independent of this study). The DHS data sets are nationally representative and regularly used in evaluating national programmes or project. Our analysis took clustering and stratification into consideration and applied sample weighting and also adjusted for confounding to minimise internal bias. The study findings of rising stillbirths could perhaps be attributed to increased utilization which may have affected quality of care, and this was corroborated by the qualitative study participants during the IDIs and FGDs.

17. Centrally to what is known, the education level in this study seem to disfavour the outcome variables (that the higher one’s education level the more likely they are to have negative outcomes). This needs to be interrogated too in the context of national maternal and perinatal mortality service delivery.

Response: We agree with the reviewers that education appears a disincentive as per the current findings. However, education as reported in this study is a confounding variable. Perhaps, analysing education as the main outcome variable could yield different results, but that is outside the scope of the current study.

18. Table 7: group discussion the unit is the focus group. Kindly indicate it and in bracket show the total number of participants in those FGDs which is 44. Revise as appropriate.

Response: This has been revised appropriately in Table 7.

19. “Antepartum (macerated) stillbirths” these happen before the onset of labour and any stillbirth happening after the onset of labour whether outside the health facility is fresh stillbirth. Revise the sentence as appropriate to reflect that death before arriving at the facility is not the cut off to determine macerated stillbirth.

Response: This has been revised accordingly in paragraph two, page 27.

20. “The reasons for the delay in arriving in the health facility”. What I infer from this and subsequent statements is that the pathway to seeking delivery services has a number of bottlenecks which include actors (Traditional birth attendants) and negative cultural practices (use of local herbs) this needs to come out clearly other than amplifying the “kaligutiem”

Response: Yes, cultural practices such as herbal intake affects the pathway to seeking care during labor. However, our use of the term ‘kalgutiem’ has been chosen to reflect the local context and observations by the service providers. However, this has been revised in pages 29 and 30.

21. The effects of data capture may need to be brought into the picture here. The increased number/rates of stillbirth may have been due to improvements in data capture that almost all cases are captured. Needs to be discussed if you think it may have an effect on the observed trends between the two timepoints.

Response: The effect of data capture was brought out by the service providers during the IDIs as reasons for the high figures. The current study did not independently analyse data capture impact to ascertain its influence on the totality of the stillbirth rate. Our analysis of the outcomes of stillbirth based on the secondary data obtained from the DHS programme.

22. “Both stillbirth and perinatal deaths were prevalent among” did you try to explore the role of data capture this might play on the observed outcomes? It may be that proper data capture in the free policy group will show many cases which may be the opposite in the other group.

Response: Although this study did not independently analyse the role of data capture, the service providers did mention ‘data capture’ as influencing the high figures during the IDIs, and this has been reported as part of the qualitative findings in page 27, paragraph two and also reflected in the discussion, paragraph nine (page 36).

23. “The findings imply that stillbirth failed to show a decline in 2014”. You may need to highlight the limitations of this study wherein the paper objectives were never the main aim of the policy so there is a likelihood that it may have affected the observed trends.

Response: This comment has been addressed in paragraph one under ‘strength and limitation’.

SIGNED 

John AZAARE

(Corresponding Author)

---

## [Editor Report · Decision Letter 3]

31 Aug 2022

Evaluating the impact of maternal health care policy on stillbirth and perinatal mortality in Ghana: a mix method approach using two rounds of Ghana demographic and health survey data sets and qualitative design technique

PONE-D-21-35524R3

Dear Dr. Azaare,

We’re pleased to inform you that your manuscript has been judged scientifically suitable for publication and will be formally accepted for publication once it meets all outstanding technical requirements.

Kind regards,

Angela Lupattelli, PhD

Academic Editor

PLOS ONE

---

## [Editor Report · Acceptance letter]

19 Sep 2022

PONE-D-21-35524R3 

Evaluating the impact of maternal health care policy on stillbirth and perinatal mortality in Ghana; a mixed method approach using two rounds of Ghana demographic and health survey data sets and qualitative design technique. 

Dear Dr. Azaare:

I'm pleased to inform you that your manuscript has been deemed suitable for publication in PLOS ONE. Congratulations! Your manuscript is now with our production department. 

Kind regards, 

on behalf of

Dr. Angela Lupattelli 

Academic Editor

PLOS ONE